**PLOS | ONE**

# Assessment of the clinical utility of four NGS panels in myeloid malignancies. Suggestions for NGS panel choice or design

**Almudena Aguilera-Diaz**[1,2], **Iria Vazquez**[2,3], **Beñat Ariceta**[3], **Amagoia Mañú**[3], **Zuriñe Blasco-Iturri**[3], **Sara Palomino-Echeverría**[3], **María José Larrayoz**[2,3], **Ramón García-Sanz**[4], **María Isabel Prieto-Conde**[4], **María del Carmen Chillón**[4], **Ana Alfonso-Pierola**[5], **Felipe Prosper**[1,2,5], **Marta Fernandez-Mercado**[1,3,6]*, **María José Calasanz**[2,3,7]

1 Advanced Genomics Laboratory, Hemato-Oncology, Center for Applied Medical Research (CIMA), University of Navarra, Pamplona, Spain, 2 Navarra Institute for Health Research (IdiSNA), Pamplona, Spain, 3 Hematological Diseases Laboratory, CIMA LAB Diagnostics, University of Navarra, Pamplona, Spain, 4 Hematology Department, University Hospital of Salamanca, IBSAL and CIBERONC, Salamanca, Spain, 5 Hematology Department, Clinica Universidad de Navarra (CUN), Pamplona, Spain, 6 Biomedical Engineering Department, School of Engineering, University of Navarra, San Sebastian, Spain, 7 Scientific Co-Director of CIMA LAB Diagnostics, CIMA LAB Diagnostics, University of Navarra, Pamplona, Spain

* mfmercado@unav.es, marfermer@yahoo.es

**Data Availability Statement:** All relevant data are within the paper and its Supporting Information files.

## Abstract

The diagnosis of myeloid neoplasms (MN) has significantly evolved through the last few decades. Next Generation Sequencing (NGS) is gradually becoming an essential tool to help clinicians with disease management. To this end, most specialized genetic laboratories have implemented NGS panels targeting a number of different genes relevant to MN. The aim of the present study is to evaluate the performance of four different targeted NGS gene panels based on their technical features and clinical utility. A total of 32 patient bone marrow samples were accrued and sequenced with 3 commercially available panels and 1 custom panel. Variants were classified by two geneticists based on their clinical relevance in MN. There was a difference in panel's depth of coverage. We found 11 discordant clinically relevant variants between panels, with a trend to miss long insertions. Our data show that there is a high risk of finding different mutations depending on the panel of choice, due both to the panel design and the data analysis method. Of note, *CEBPA*, *CALR* and *FLT3* genes, remains challenging the use of NGS for diagnosis of MN in compliance with current guidelines. Therefore, conventional molecular testing might need to be kept in place for the correct diagnosis of MN for now.

## Introduction

Myeloid neoplasms (MN) comprise a group of clonal disorders biologically and clinically heterogeneous characterized by ineffective hematopoiesis, due to Hematopoietic Stem Cells (HSC) excessive proliferation and defective myeloid linage differentiation [1].

**Funding:** This work was supported by CIMA LAB Diagnostics research program and grants from the Spanish Association against Cancer (AECC, AIO2014), Government of Navarra, Department of Industry, Energy and Innovation (Project DIANA, 0011-1411-2017-000028) (FP, IV, MJL, MFM, MJC, http://www.proyectodiana.es/), and Instituto de Salud Carlos III (PI16/00159 and PI17/00701) (MFM, AAD, FP, https://www.isciii.es/Paginas/Inicio.aspx). AAD is supported by a CIMA´s fellowship, and IV is supported by Pethema Foundation. The funders had no role in study design, data collection and analysis, decision to publish, or preparation of the manuscript.

**Competing interests:** The authors have declared that no competing interests exist.

The diagnosis of myeloid malignancies has significantly evolved through the last few decades. Nowadays, blood cell morphology, blast count, cytogenetics and molecular analysis are crucial for clinicians to diagnose and to predict prognosis of MN following the World Health Organization (WHO) classification [2]. This classification includes the genetic characterization of genes such as *JAK2*, *MPL* and *CALR* for Myeloproliferative Neoplasms (MPN); *ASXL1*, *CEBPA*, *DNMT3A*, *FLT3*, *IDH1/2*, *KIT*, *KMT2A*, *NPM1*, *RUNX1*, *TET2*, *TP53* and *WT1* genes for Acute Myeloid Leukemia (AML); and *SF3B1*, for Myelodysplastic Syndromes (MDS). Along the last few years, the scientific community has deepened its understanding on the genetic aberration associated to MN through the discovery of other recurrently mutated genes such as *ASXL1*, *DNMT3A*, *EZH2*, *RUNX1*, *SRSF2*, *TET2*, *TP53* and *U2AF1* in MDS [3] [4], and *ASXL1*, *CBL*, *EZH2*, *NRAS/KRAS*, *RUNX1*, *SETBP1*, *SRSF2* and *TET2* in Chronic Myelomonocytic Leukemia (CMML) [5][6][7][8]. A number of these genes have been related to patient prognosis; for example, it is well known that mutations in *SF3B1* gene in MDS with ring sideroblasts (MDS-RS) are related to good prognosis [9], whereas mutations in *TP53* gene are usually related to poor outcomes [10]. These discoveries are crucial to help clinicians in the management of the disease, hence the correct characterization of the genes is vital.

Hematological malignancies are genetically heterogeneous, and recent studies have elucidated the importance of genomic testing (rather than individual gene testing) to understand the pathology of the disease [3][4][11]. Due to its wide scope, Massive Parallel Sequencing (also called Next Generation Sequencing, NGS) is being increasingly used for genomic characterization of clinical samples. NGS is nowadays not just an essential tool for the discovery of new gene mutations, but is also becoming a rather useful technique to improve patient diagnosis, prognosis and treatment based on identified tumor variants.

There are several ways to perform NGS on DNA, including whole-genome sequencing (WGS), which allows sequencing of the entire genome; whole-exome sequencing (WES), which focuses on the coding regions (exons), encompassing ~2.5% of the total human genome; and targeted sequencing (also known as NGS panels), which focuses on a certain number of genes, generally involved in the biology of a specific disease [12]. NGS panels are the NGS tools most widely used for clinical applications, mainly for cost effectiveness reasons, but also because they allow deeper sequencing, permitting detection of small mutant clones. For MN there is a plethora of different NGS panels developed by research groups all over the world as well as commercially available panels.

In this study we have compared the analytic performance of four NGS panels focused on myeloid malignancies. To that end, samples from 32 patients with MN were sequenced using three different commercially available targeted gene panels, offered by Illumina, Oxford Gene Technology (OGT), and SOPHiA GENETICS; the other one is a customized pan-myeloid panel developed in collaboration with SOPHiA GENETICS. The aim of this study is to dissect a number of NGS panels available for genomic characterization of MN, discuss their design, chemistry, analysis pipeline, and whether they cover and detect mutations in the most relevant genes related to MN. We hope to offer helpful criteria to hematological genetic laboratories when implementing new NGS panels.

## Materials and methods

### Patient samples

A total of 32 patient bone marrow (BM) samples were accrued: 17 with AML, 7 with MPN, 6 with MDS, and 2 with CMML. BM was the tissue of choice for analysis following European recommendations [13]. Seventeen of those samples were analyzed with TruSight™ Myeloid Panel (TSMP) (Illumina, San Diego, CA, USA), 16 with SureSeq™CoreMPN Panel and

SureSeq™AML Panel (SureSeq) (Oxford Gene Technology, Oxford, UK), 15 with Myeloid Solutions ™panel (MYS) (SOPHiA GENETICS, Saint Sulpice, Switzerland), and all 32 were tested with a custom Pan-Myeloid Panel (PMP) (University of Navarra and University Hospital of Salamanca) (Fig 1).

All DNA samples were extracted using QIAamp DNA Blood Mini Kit (Qiagen, Hilden, Germany), quantified using Qubit dsDNA BR Assay Kit on a Qubit 3.0 Fluorometer (Life Technologies, Carlsbad, CA, USA), and DNA quality was assessed by DNA genomic kit on a Tape Station 4100 (Agilent Technologies, Santa Clara, CA, USA).

DNA samples from 15 patients were sent to SOPHiA GENETICS (Saint Sulpice, Switzerland) and 16 DNA samples to Oxford Gene Technology (OGT) (Oxford, UK) for library preparation, sequencing, and variant calling.

Samples and data from patients included in the study were provided by the Biobank of the University of Navarra (UN) and were processed following standard operating procedures approved by the CEI (Comité de Ética de la Investigación) of UN. Patient's data were fully anonymized, and all patients provided informed written consent to have data from their medical records such as age, gender and diagnosis to be used for research purposes.

### TruSight Myeloid Panel (TSMP)

TruSight Myeloid Panel (TSMP) (Illumina, San Diego, CA, USA), consists of 568 amplicons of 250 base pairs (bp) in length, with a total genomic footprint of 141 kb, targeting the full CDS of 15 genes and exonic hot spots of 39 additional genes (Fig 2) (S1 Table).

Libraries of 17 patient's samples were prepared by our team following manufacturer's instructions. Libraries quality was assessed using DNA D1000 kit and a Tape Station 4100 (Agilent Technologies, Santa Clara, CA, USA), and libraries quantity was assessed with Qubit dsDNA HS Assay Kit and Qubit 3.0 Fluorometer (Life Technologies, Carlsbad, CA, USA). Libraries were normalized according to the measured quantity and pooled together at 4nM.

A total of 10.5 pM of the 8 pooled libraries was pair-end sequenced on a MiSeq (Illumina, San Diego, CA, USA) with 201x2 cycles using the Reagent Kit V3 600 cycles cartridge, according to manufacturer's instructions. Bam and Variant Calling Files (VCF) were directly obtained from MiSeq instrument and variants were annotated using Variant Studio (Illumina, San Diego, CA, USA).

### Myeloid Solutions™ Panel

Myeloid Solutions™ Panel (MYS) (SOPHiA Genetics, Saint Sulpice, Switzerland), consists in a hybridization capture-based panel, with a total genomic footprint of 49 kb, targeting the full CDS of 10 genes and exonic hotspots of 20 additional genes (Fig 2) (S2 Table).

Extracted DNA from 15 patient samples was sent to SOPHiA GENETICS facilities, where they carried out libraries preparation and pair-end sequencing on a MiSeq (Illumina, San Diego, CA, USA) with 251x2 cycles using Reagent Kit V3 600 cycles cartridge, according to manufacturer´s instructions. Alignment, base calling and variant annotation were performed with SOPHiA DDM software.

### SureSeq™ panels

SureSeq™ AML Panel and SureSeq™ Core MPN Panel (Oxford Gene Technology, Oxford, UK), consists in 2 hybridization capture-based panels with a total genomic footprint covering 53 kb; one panel targets the full CDS of 20 genes, and the other one targets exonic hotspots of 3 additional genes (*MPL*, *JAK2* and *CALR*) (Fig 2) (S3 Table).

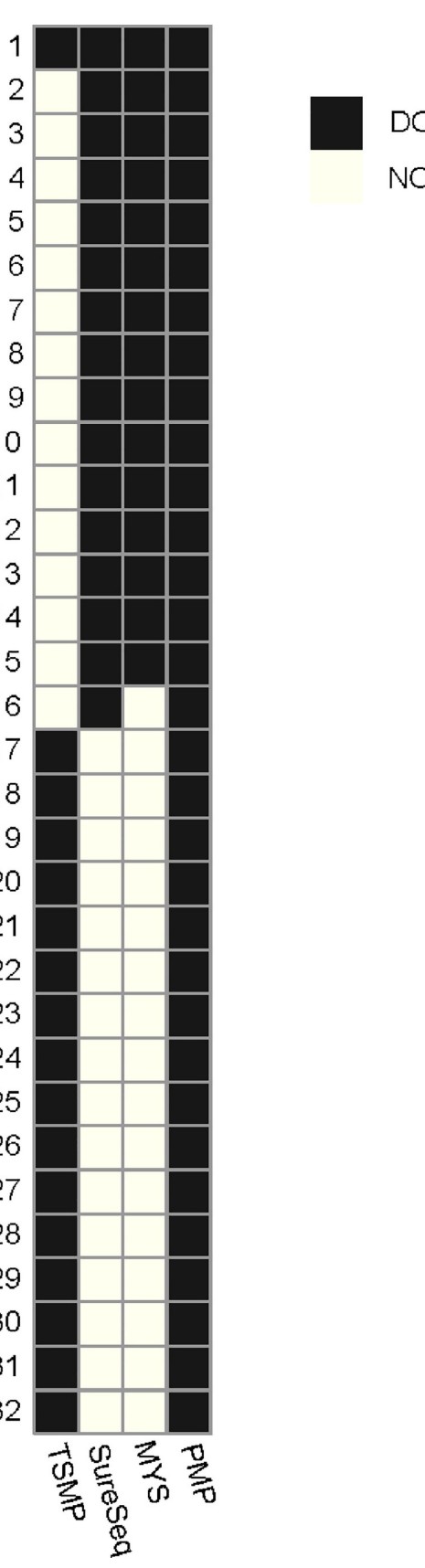

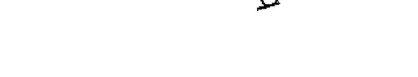

**Fig 1. Samples assessed by each panel.** Thirty two bone marrow patient samples (17 AML, 7 MPN, 6 MDS, and 2 CMML) were sequenced: 17 were assessed with TSMP (Illumina, San Diego, CA, USA), 16 with SureSeq panels (Oxford Gene Technology, Oxford, UK) panel, 15 with MYS panel (SOPHiA GENETICS, Saint Sulpice, Switzerland) panel, and all 32 were tested with the custom PMP.

Extracted DNA from the same 15 patients sent to SOPHiA GENETICS, was sent to OGT facilities, where they carried out library preparation according to their own protocol. Libraries were pair-end sequenced on a MiSeq (Illumina, San Diego, CA, USA) with 151x2 cycles using Reagent Kit V2 cartridge, according to manufacturer´s instructions.

## Pan-Myeloid Panel (PMP)

Pan-Myeloid Panel (PMP) consists in a hybridization capture-based panel developed by the UN (Pamplona, Spain) and the University Hospital of Salamanca (Salamanca, Spain) in collaboration with SOPHiA GENETICS (Saint Sulpice, Switzerland). It counts on a total genomic footprint of 114 kb, targeting 63 genes. For the detection of Single Nucleotide Variants (SNV), insertions and deletions (indels) we targeted 48 genes: full CDS of 22 genes, and exonic hotspots of 26 additional genes (Fig 2) (S4 Table). This panel was also designed with the aim of detecting Copy Number Variations (CNV) in chromosomes 5, 7, 8 and 20; these data have not been included in the present study.

Libraries were carried out following manufacturer's instructions. Final libraries quantity was measured using the Qubit dsDNA HS Assay Kit in a Qubit 3.0 Fluorometer (Life Technologies, Carlsbad, CA, USA), and libraries quality was assessed using DNA D1000 kit, and visualized on the Agilent 4100 Tape Station (Agilent Technologies, Santa Clara, CA, USA). Libraries were normalized and pooled together at 4nM.

A total of 10.5 pM of 8 pooled libraries was pair-end sequenced on the MiSeq (Illumina, San Diego, CA, USA) with 251x2 cycles using the Reagent Kit V3 600 cycles cartridge, according to manufacturer's instructions. Raw data were directly obtained from the MiSeq and uploaded onto SOPHiA DDM software, where alignment, variant calling and annotation were performed.

## Sequencing and variant data analysis

Aligned reads were counted using SAMTools version 1.6. Read counting and plotting were performed using R version 3.4.2 (RStudio, Boston, MA, USA).

SureSeq™ panels bam files analysis was performed using VarScan version 2.3.9, with strand bias filters and setting minimum read to 5. Variant calling of the other three panels was performed within SOPHiA DDM software version 5.2.7.1 (SOPHiA GENETICS, Saint Sulpice, Switzerland) for MYS and PMP, or within the MiSeq (Illumina, San Diego, CA, USA) for TSMP.

List of annotated variants were reviewed for filtering out of intronic, intergenic and splice regions variants. Only variants with a minimum variant allele frequency (VAF) of 5% and with a minimum coverage of 100 reads were kept to avoid potential sequencing errors. Variants were categorized by two geneticists with expertise in hematological malignancies, and only variants classified as pathogenic and likely pathogenic were considered clinically relevant. Clinical classification of the variants was individually reviewed according to current guidelines from the Spanish Group of Myelodysplastic Syndromes [14]. Aligned reads were manually curated for confirmation of the presence of the filtered-in variants within the Integrative Genomics Viewer (IGV) software (Broad Institute) [15]. Variant data were summarized using

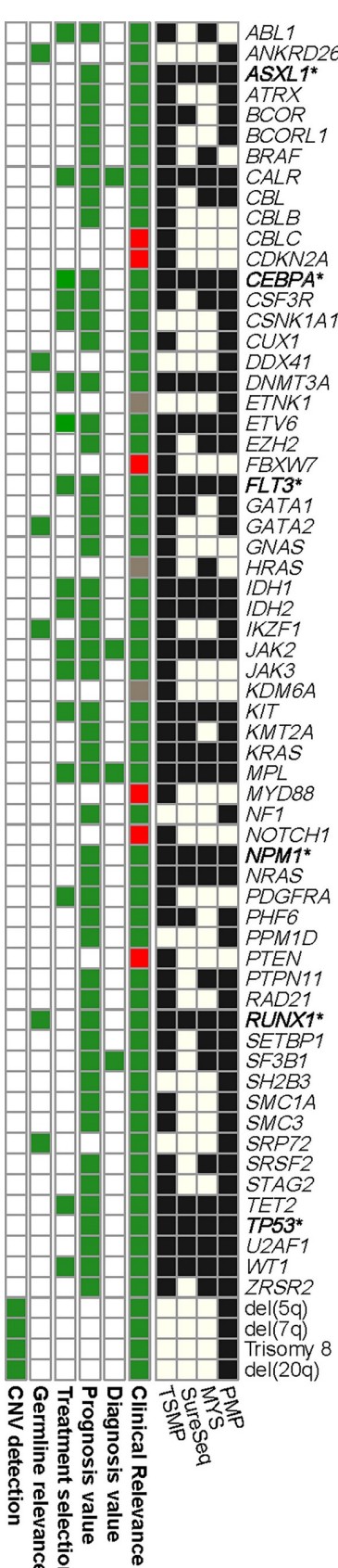

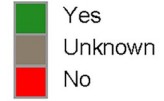

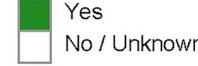

**Fig 2. Genes covered by each panel and their clinical relevance.** The 62 genes included in the present study are listed on the right. Black color denotes which gene is covered in each panel. Green color highlights the 53 genes that have been described as clinically relevant for MN, since they show diagnostic, prognostic and/or predictive value, or they have been related to predisposition to develop MN. Red color represents genes that are not clinically relevant in MN. Grey color marks those genes that has been described in MN but their clinical relevance is still unknown.

median and range, and plotted using GraphPad Prism 5 (GraphPad, La Jolla, California, USA).

## Genetic molecular testing

Purity and concentration of the extracted DNA were measured using a NanoDrop 1000 spectrophotometer (ThermoFisher SCIENTIFIC, Waltham, MA, USA).

Mutations in *CEBPA* exon were detected by genomic DNA PCR, cloning and Sanger sequencing using the primers and following the procedures previously described [16][17]. Mutations in *CALR* exon 9 were assessed by PCR and Sanger sequencing [18]. *FLT3* exons 14 and 15 were assessed by PCR and capillary electrophoresis using 5ng of genomic DNA per samples to detect the presence of internal tandem duplications (ITD) [19]. The ratio of *FLT3*-ITD to wild-type *FLT3* was quantified by the Applied Biosystems sequencing software GeneScan® as described previously [20]. *FLT3* exon 20 was tested by PCR and RFLP analysis for presence of mutations in codons p.Asp835/p.Ile836 [21]. PCR products were Sanger sequenced at Macrogen Europe´s facilities (Amsterdam, Netherlands).

The molecular analysis data obtained by conventional molecular techniques for all patients are shown in Table 1. Patients 1, 5 and 8 harbored biallelic *CEBPA* mutations; patients 2, 3, 7 and 12 harbored *FLT3*-ITD favorable ratio (< 0.5) and *NPM1* not mutated; patient 11 had *FLT3*-ITD favorable ratio and mutated *NPM1*; patients 4, 9, 10 and 13 presented monoallelic *CEBPA*; patients 6 and 14 had *CALR* mutated; patients 15 and 16 had unfavorable *FLT3*-ITD ratio (> 0.5); and patient 23 presented triple negative MPN (*CALR*, *JAK2* and *MPL* genes non mutated). The 14 remaining patients had not been tested by conventional molecular techniques.

## Results

### Comparison of the NGS panels characteristics

**a) Panels performance.** Based on the technology used for capturing the genomic regions of interest for library preparation there are two types of NGS targeted panels: hybridization capture-based libraries or amplicon-based libraries. TSMP was the only amplicon-based panel in this study; the other three panels (SureSeq, MYS and PMP) were hybridization capture-based panels. Library preparation for TSMP and SureSeq panels took one day, whereas for PMP and MYS panel took two working days. All panel's chemistry was compatible with the Illumina sequencer MiSeq, but differ in the sequencing time, due to the number of sequencing cycles: PMP took the longest run time (50h, 250x2 cycles) and SureSeq panels the shorter run time (less than 24h, 151x2 cycles). Software analysis were available for TSMP, PMP and MYS panels at the time of the study. The performance of the panels is summarized in Table 2.

**b) Panels design and clinical relevance of the genes covered.** All four panels analized de same 19 genes (core myeloid gene set), among others, those being *ASXL1*, *CALR*, *CEBPA*, *DNMT3A*, *ETV6*, *FLT3*, *IDH1*, *IDH2*, *JAK2*, *KIT*, *KRAS*, *MPL*, *NPM1*, *NRAS*, *RUNX1*, *TET2*, *TP53*, *U2AF1*, *WT1* (Figs 2 and 3). However, the target regions for that core myeloid gene set differ between the four panels included in this study (S1 Fig). Panels design and clinical relevance of the genes are represented in Fig 2.

**Table 1. Conventional molecular testing data of patients included in the study.**

| Patient ID | Pathology | Karyotype | FISH | Molecular |
|---|---|---|---|---|
| 1 | AML | 46, XY [30] | NP | *CEBPA* biallelic |
| 2 | AML secondary to MDS | 46,XX, del(20)(q12)[15]/46,XX[15] | NP | *FLT3*-ITD favorable/*NPM1* non mutated |
| 3 | AML secondary to treatment | 46,XX del(11)t(11;11)(p15;q23)[23]/46,XX[7] | 11q23 (*KMT2A/MLL*) negative | *FLT3*-ITD favorable/*NPM1* non mutated |
| 4 | AML | null | *RUNX1-RUNXT1* negative | *CEBPA* monoallelic |
| 5 | AML M1 | NP | *PDGFRβ*, *FGFR1* negative | *CEBPA* biallelic |
| 6 | Essential Thrombocytopenia | NP | NP | *CALR* |
| 7 | AML M5 | NP | NP | *FLT3*-ITD favorable/*NPM1* non mutated/*WT1* overexpressed |
| 8 | AML | NP | NP | *CEBPA* biallelic |
| 9 | AML M1 | NP | *PDGFRβ* negative | *CEBPA* monoallelic /*FLT3* non mutated |
| 10 | AML | 46, XY [30] | NP | *CEBPA* monoallelic |
| 11 | AML M1 | NP | NP | *FLT3*-ITD favorable/*NPM1* mutated |
| 12 | AML | NP | NP | *FLT3*-ITD favorable/*CEBPA* and *NPM1* non mutated |
| 13 | AML | 46, XX [30] | NP | *CEBPA* monoallelic |
| 14 | Essential Thrombocytopenia | NP | NP | *CALR* mutated /*JAK2* non mutated |
| 15 | AML secondary CMML | Null | NP | *FLT3*-ITD (ratio 1,11) Unfavorable |
| 16 | AML | 46, XY [30] | NP | *FLT3*-ITD (ratio 1,06) Unfavorable |
| 17 | MDS | 45,X,-Y[29]/46,XY[1] | del(5q) and del (7q) negative | NP |
| 18 | AML M2 | NP | NP | NP |
| 19 | MDS | 47,XY,+13[10]/46,XY[40] | del(5q), del (20q) and del (7q) negative | NP |
| 20 | MDS-EB1 | 46,XX [30] | del(5q), del (20q) and del (7q) negative | NP |
| 21 | Myelofibrosis | NP | NP | NP |
| 22 | Myelofibrosis | NP | NP | NP |
| 23 | Myelofibrosis | Null | NP | MPN Triple Negative |
| 24 | CMML | 46,XX [30] | NP | NP |
| 25 | MDS | 46,XX [30] | del(5q), del (20q) and del (7q) negative | NP |
| 26 | Polycythemia Vera | NP | NP | NP |
| 27 | Myelofibrosis | NP | NP | NP |
| 28 | MDS (del(5q)) | NP | NP | NP |
| 29 | AML | NP | NP | *FLT3* (ITD—D835) non mutated/*CEBPA* and *NPM1* non mutated |
| 30 | AML in treatment | 46,XY,t(3;6)(q26;q21) | NP | NP |
| 31 | MDS-EB2 | 46,XY,inv(9)(p12q13)[30] | NP | NP |
| 32 | CMML | 46,XY,add(15)(p13),add(21)(q22) [30] | NP | NP |

AML = Acute Myeloid Leukemia; NP = Non Performed; MDS = Myelodisplastic Syndromes; CMML = Chronic Myelomonocytic Leukemia; MDS-EB = Myelodisplastic Syndromes with Excess Blasts; MPN = Myeloproliferative Neoplasm

For example, exon 10 of *MPL* gene is included in all panels, whereas exons 3–6 and 12 are targeted only by PMP. Similarly, *ASXL1* exon 12 is covered by all panels, while SureSeq™ AML covers *ASXL1* full CDS (S2 Fig, S1–S4 Tables).

**Table 2. Characteristics of panel performance.**

|  | PMP (SOPHiA GENETICS) | MYS (SOPHiA GENETICS) | SureSeq (OGT) | TSMP (Illumina) |
|---|---|---|---|---|
| **Number of samples** | 32 | 15 | 16 | 17 |
| **Type of library preparation** | Hybridization capture | Hybridization capture | Hybridization capture | Amplicon-based |
| **Wet-lab working time (days)** | 2 | 2 | 1 | 1 |
| **Possibility of customization** | Yes | Yes | Yes | No |
| **Sequencing cycles and time** | 251cycles/50h | 251cycles/48h | 151cycles/24h | 201cycles/40h |
| **Analysis Software** | SOPHiA DDM | SOPHiA DDM | Under development at the time of the study | Variant Studio |

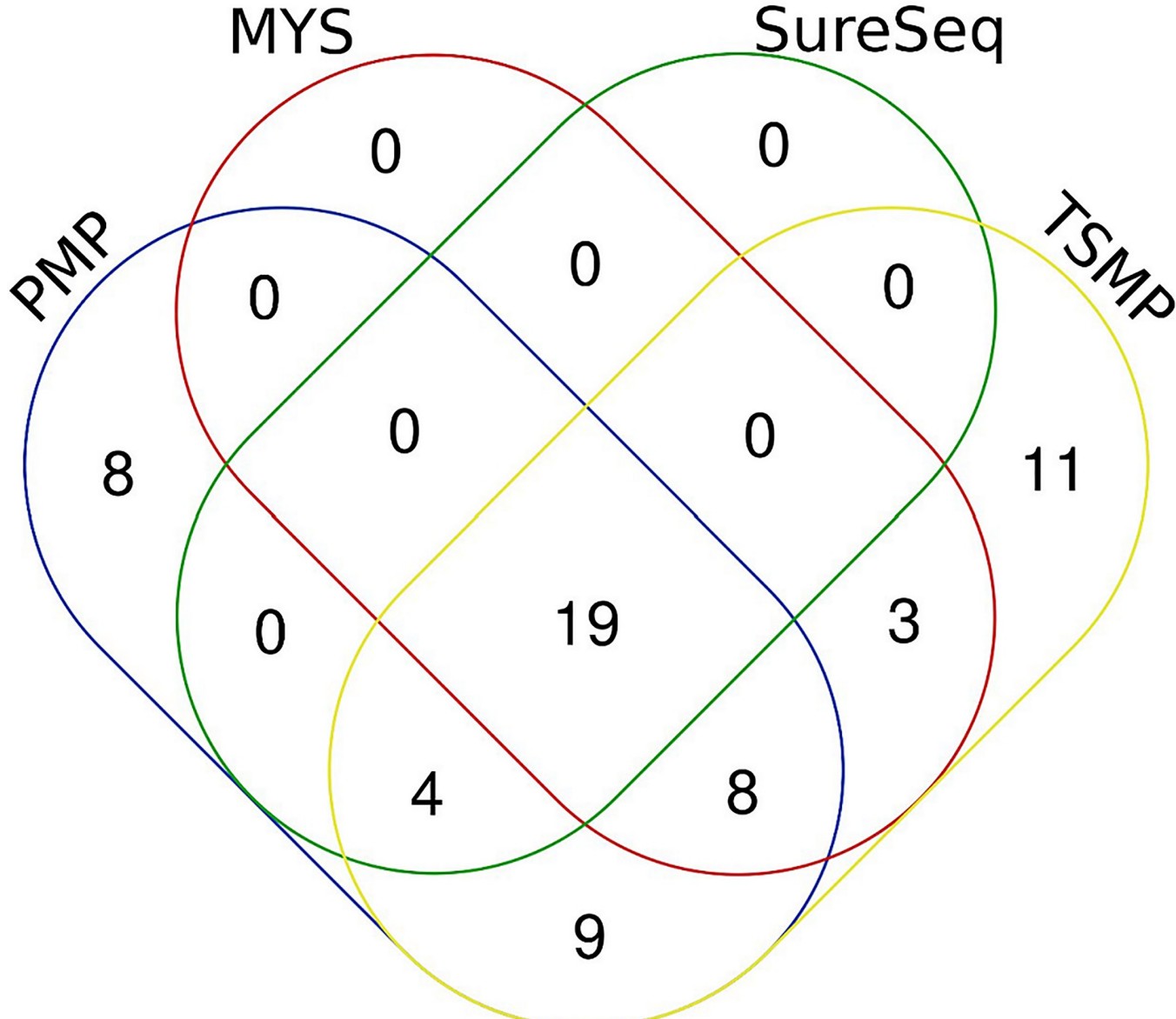

**Fig 3. Number of genes shared between panels.** All four panels covered the same 19 genes (core myeloid gene set). TSMP, PMP and Sureseq panels design includes 4 genes not targeted by MYS. PMP, TSMP and MYS panels target 8 genes not included in SureSeq panel design. TSMP and PMP cover 9 genes that are not within MYS and SureSeq panel scope. TSMP and MYS panels cover 3 genes not included in the other two panels.

The 19 genes included in the core myeloid gene set, have extensively been described as relevant in different myeloid malignancies. All of them show prognostic value; *CALR*, *JAK2* and *MPL* have also diagnostic value; and *CALR*, *DNMT3A*, *JAK2*, *KIT*, *FLT3*, *IDH1/2* and *TET2* have been shown to bear predictive value. The remaining genes included in the panels, fine tune the design so they were useful for different aims. For example, SureSeq™ panels were designed for analysis of AML and MPN cases, but it lacked essential genes for the study of MDS, such as genes involved in splicing (*SF3B1*, *SRSF2*, *ZRSR2*), epigenetic regulation (*EZH2*), transcriptional regulation (*GATA2*) or signal transduction (*CBL*) [22][23][24][25]. Similarly, MYS panel was designed to characterize the mutational landscape of MDS, MPN and AML, but it missed a number of relevant genes such as the transcription regulators *GATA2*, *IKZF1*, and *PHF6* [25][26]. On the contrary, TSMP included some genes relevant to lymphoid malignancies, such as *MYD88*, *NOTCH1* and *PTEN* [27][28][29]. In addition, PMP was the only one that included the analysis of myeloid-relevant genes as *CSNK1A1*, *NF1*, *PPM1D*, and *SH2B3* [30][31][32][33]. However, there is still room for PMP improvement, because it lacked targeting the recently described mutated exons in *FLT3* gene [34], which are covered only by SureSeq™ AML panel. The recurrence of mutations for different MNs in the genes covered in any of the analyzed panels is summarized in S5 Table.

## Comparison of the NGS panels coverage

Depth of coverage is the average number of mapped reads at a given locus in the panel. The importance of a good panel coverage resides in the fact that a low coverage limits the ability to confidently call a variant present in the sample, especially those variants with low allele frequency. Fig 4 shows the mean of depth of coverage for each panel by gene; a mean coverage of 1000x allows detection of clones present at 0.1% (cut-off value of 10 reads, assuming there is no strand-bias).

All panels showed mean coverage over 1000x. However, we observed that TSMP did not cover *CEBPA* gene as homogeneously as the other panels; this might be because TSMP is an amplicon-based panel, and *CEBPA* is a one-exon gene lying within a CpG Island [20]. Therefore, PCR-based library preparation struggles to amplify (and capture) this gene, challenging the detection of variants in *CEBPA* gene (S3 Fig). S4 Fig shows the mean coverage by region targeted for each panel.

## Comparison of the detected variants in all four NGS panels

Filtered VCF obtained from the different software (from SOPHiA GENETICS and Illumina) and the in-house analysis of the SureSeq panels from all samples were compared. The number of variants called in each panel is plotted in Fig 5, and the VAFs comparison is represented in S5 Fig.

**a) Comparison of all coding variants detected.**

i. Called coding variants. A total of 1146 coding variants were detected by all four panels. Fig 5A shows that PMP was the panel that called a higher number of variants per patient (mean = 26) followed by TSMP (mean = 24), MYS panel (mean = 16), and SureSeq panels, which were the ones that called a lower number of variants (mean = 15). This might be due to the fact that PMP and TSMP were the larger panels, covering more genes (S1–S4 Tables).

ii. Coding variants called in the core myeloid gen set. When focusing on the core myeloid gene set of 19 genes, a total of 367 variants were detected by all four panels. SureSeq panels called a higher number of variants per patient (mean = 13), followed by MYS (mean = 9.2) and PMP (mean = 8.1); TSMP was the panel that called a lower number of variants

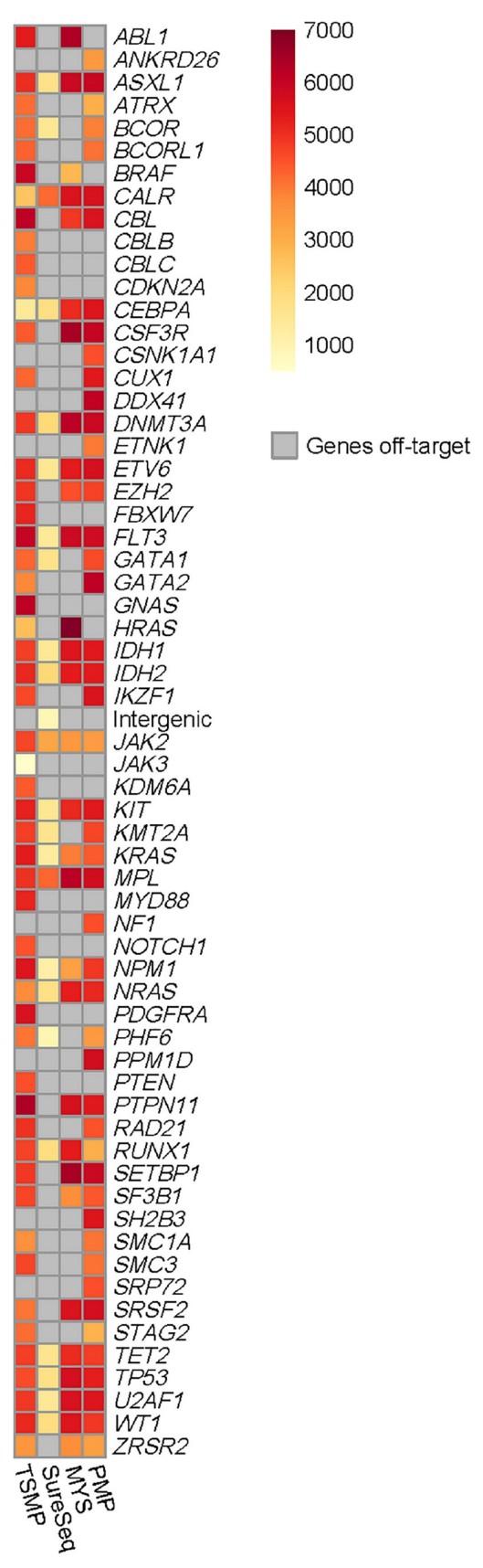

**Fig 4. Panel coverage.** The mean coverage by gene in each panel is represented in yellow (1000x) through dark red (7000x).

(mean = 7.8) (Fig 5B). Analysing in detail these differences, SureSeq panels were the ones that called more variants because it covers the whole CDS of the myeloid core gene set, and presicely *ASXL1*, *FLT3*, *IDH1*, *IDH2*, *KIT*, *KRAS*, *NPM1*, *NRAS*, *U2AF1* and *WT1* are the genes harboring more variants in our cohort (S1 Fig). Similarly, MYS panel covered the whole *JAK2* gene, whereas PMP included exons 12 to 15 only, what led MYS panel calling more variants than PMP. Finally, PMP called more variants than TSMP because it analized more exons of *MPL* gene, and TSMP struggled covering *CEBPA* gene, as mention above (S1, S2 and S4 Figs).

**b) Comparison of the clinically relevant variants detected.** Since these panels were designed with the intention of being clinically useful, we repeated the analysis, focusing on the clinical relevance of the variants called. Variants were classified by two geneticists with expertise in hematological malignancies. Variants classified as "pathogenic" or "likely pathogenic" were kept as clinically relevant. Table 3 shows all clinically relevant mutations detected in each patient.

i. Called clinically relevant variants. A total of 50 clinically relevant variants were detected by all four panels. PMP and TSMP were the panels that called a higher number of clinically

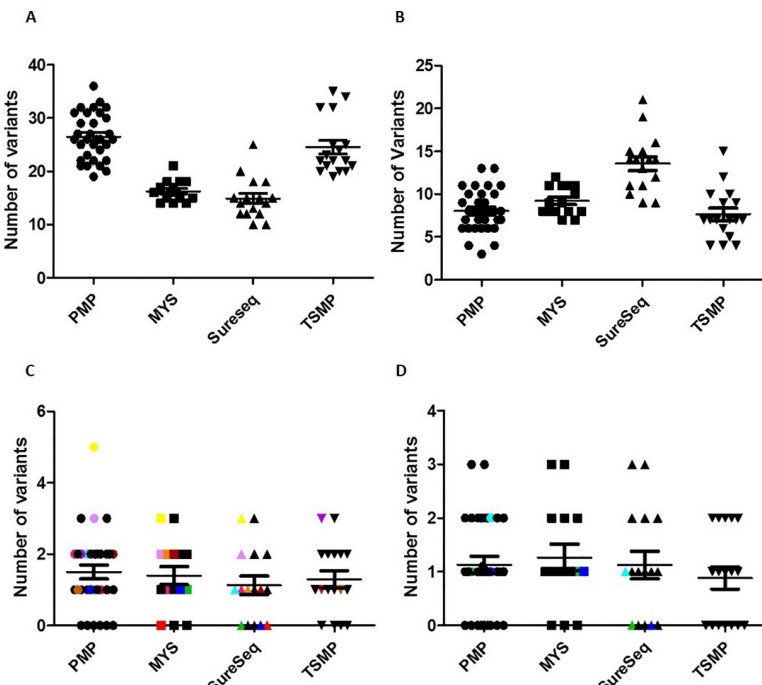

**Fig 5. Number of variants called by panel.** Each data point represents the number of variants called in each sample. **A:** Coding variants. **B:** Coding variants called in the core myeloid gene set. **C:** Clinically relevant variants. Coloured data highlight those patients with clinically relevant variants missed by any of the panels, either because those genes are not included in panel design, or because of panel issues. Each colour represents the same patient. **D:** Clinically relevant variants in the core myeloid gene set. Patients 7 (green), 14 (blue) and 16 (turquoise) are highlighted because they miss three clinically relevant mutations (one each).

**Table 3. Clinically relevant mutations detected by panel.**

| Sample ID | P | M | S | T | Gene | Chr | Position | CodCons | Transcript | c.DNA | Protein | Classification |
|---|---|---|---|---|---|---|---|---|---|---|---|---|
| 1 | C | NI | NI | C | GATA2 | 3 | 128202767 | missense | NM_001145661 | c.953C>G | p.Ala318Gly | Pathogenic/COSM249850 |
| 2 | C | C | C | ND | NRAS | 1 | 115256529 | missense | NM_002524 | c.182A>G | p.Gln61Arg | Pathogenic/COSM584 |
| 2 | C | C | C | ND | RUNX1 | 21 | 36259151 | frameshift | NM_001001890 | c.257_258delCC | p.Pro86Hisfs*24 | Likely Pathogenic |
| 2 | C | C | C | ND | TET2 | 4 | 106157572 | frameshift | NM_017628 | c.2474delC | p.Ser825*fs*1 | Likely Pathogenic/ COSM4170127 |
| 2 | C | NI | NI | ND | BCORL1 | X | 129149890 | nonsense | NM_021946 | c.3142C>T | p.Arg1048* | Likely Pathogenic |
| 2 | C | NI | NI | ND | SH2B3 | 12 | 111885286 | missense | NM_005475 | c.1174C>T | p.Arg392Trp | Likely pathogenic/COSM4384767 |
| 3 | C | C | C | ND | FLT3 | 13 | 28608281 | missense | NM_004119 | c.1775T>C | p.Val592Ala | Likely Pathogenic/ COSM19522 |
| 3 | C | C | C | ND | NRAS | 1 | 115258744 | missense | NM_002524 | c.38G>A | p.Gly13Asp | Pathogenic/COSM573 |
| 3 | C | C | NI | ND | PTPN11 | 12 | 112888162 | missense | NM_002834 | c.178G>T | p.Gly60Cys | Likely Pathogenic |
| 4 | C | C | C | ND | CEBPA | 19 | 33792381 | inframe_3 | NM_004364 | c.937_939dupAAG | p.Lys313dup | Likely pathogenic/ COSM18397/ COSM18099 |
| 4 | C | NI | NI | ND | GATA2 | 3 | 128202767 | missense | NM_001145661 | c.953C>T | p.Ala318Val | Pathogenic/COSM255084 |
| 5 | C | C | C | ND | CEBPA | 19 | 33792384 | inframe_3 | NM_004364 | c.934_936dupCAG | p.Gln312dup | Pathogenic/COSM18466 |
| 5 | C | C | C | ND | CEBPA | 19 | 33793252 | frameshift | NM_004364 | c.68delC | p.Pro23Argfs*137 | Pathogenic/COSM18544 |
| 6 | C | C | C | ND | CALR | 19 | 13054591 | frameshift | NM_004343 | c.1119delC | p.Asp373Glufs*? | Likely pathogenic |
| 7 | C | C | NC | ND | FLT3 | 13 | 28608271 | inframe_36 | NM_004119 | c.1749_1784dup | p.Phe594_Arg595ins12 | Pathogenic/ITD |
| 10 | C | C | C | ND | TET2 | 4 | 106158187 | nonsense | NM_017628 | c.3088C>T | p.Gln1030* | Likely Pathogenic/COSM4766113 |
| 11 | C | C | C | ND | FLT3 | 13 | 28608255 | inframe_21 | NM_004119 | c.1800_1801ins21 | p.Asp600_Leu601ins7 | Pathogenic/ITD |
| 11 | C | C | C | ND | IDH2 | 15 | 90631934 | missense | NM_002168 | c.419G>A | p.Arg140Gln | Pathogenic/COSM41590 |
| 11 | C | C | C | ND | NPM1 | 5 | 170837543 | frameshift | NM_002520 | c.860_863dupTCTG | p.Trp288Cysfs*? | Pathogenic/COSM17559/ COSM158604 |
| 12 | C | C | C | ND | FLT3 | 13 | 28608273 | inframe_33 | NM_004119 | c.1782_1783ins33 | p.Phe594_Arg595ins11 | Pathogenic/ITD |
| 12 | C | C | C | ND | RUNX1 | 21 | 36252865 | missense | NM_001001890 | c.416G>A | p.Arg139Gln | Likely pathogenic/COSM6908427 |
| 13 | C | C | C | ND | TET2 | 4 | 106157416 | nonsense | NM_017628 | c.2317G>T | p.Gly773* | Likely pathogenic |
| 13 | C | C | NI | ND | SRSF2 | 17 | 74732959 | missense | NM_001195427 | c.284C>A | p.Pro95His | Pathogenic/COSM211504 |
| 14 | C | C | NC | ND | CALR | 19 | 13054564 | frameshift | NM_004343 | c.1099_1150del | p.Leu367Thrfs*46 | Pathogenic/ COSM1738055 |
| 15 | C | C | C | ND | DNMT3A | 2 | 25470908 | nonsense | NM_022552 | c.853G>T | p.Glu285* | Likely pathogenic/COSM4383607 |
| 15 | C | C | C | ND | FLT3 | 13 | 28608256 | inframe_24 | NM_004119 | c.1799_1800ins24 | p.Asp600_Leu601ins8 | Pathogenic/ITD |
| 15 | C | C | NI | ND | SF3B1 | 2 | 198267359 | missense | NM_012433 | c.1998G>T | p.Lys666Asn | Pathogenic/COSM131557 |
| 16 | C | ND | NC | ND | FLT3 | 13 | 28608286 | inframe_36 | NM_004119 | c.1734_1769dup | p.Tyr589_Phe590ins12 | Pathogenic/ITD |
| 16 | C | ND | C | ND | WT1 | 11 | 32417924 | frameshift | NM_000378 | c.1076_1077insT | p.Thr360Aspfs*8 | Likely pathogenic |
| 17 | C | ND | ND | C | U2AF1 | 21 | 44514777 | missense | NM_001025203 | c.470A>C | p.Gln157Pro | Pathogenic/COSM211534 |
| 18 | C | ND | ND | C | NRAS | 1 | 115258747 | missense | NM_002524 | c.35G>A | p.Gly12Asp | Pathogenic/COSM564 |
| 18 | C | ND | ND | C | SRSF2 | 17 | 74732960 | missense | NM_001195427 | c.283C>A | p.Pro95Thr | Pathogenic/COSM307353 |
| 18 | C | ND | ND | C | WT1 | 11 | 32417911 | frameshift | NM_000378 | c.1089_1090insGCCCTCTTGTACGG | p.Ser364Alafs*73 | Likely pathogenic |
| 20 | NI | ND | ND | C | GNAS | 20 | 57484421 | missense | NM_080425.2 | c.2531G>A | p.Arg844His | Likely pathogenic/COSM94388 |
| 20 | C | ND | ND | C | ASXL1 | 20 | 31024704 | missense | NM_015338 | c.4189G>A | p.Gly1397Ser | Pathogenic/COSM133033 |

*(Continued)*

**Table 3.** (Continued)

| Sample ID | P | M | S | T | Gene | Chr | Position | CodCons | Transcript | c.DNA | Protein | Classification |
|---|---|---|---|---|---|---|---|---|---|---|---|---|
| 20 | C | ND | ND | C | SF3B1 | 2 | 198267484 | missense | NM_012433 | c.1873C>T | p.Arg625Cys | Pathogenic/COSM110696 |
| 21 | C | ND | ND | C | CALR | 19 | 13054564 | frameshift | NM_004343 | c.1099_1150del | p.Leu367Thrfs*46 | Pathogenic/ COSM1738055 |
| 23 | C | ND | ND | C | MPL | 1 | 43815009 | missense | NM_005373 | c.1544G>T | p.Trp515Leu | Pathogenic/COSM3719407, COSM18918 |
| 24 | C | ND | ND | C | SETBP1 | 18 | 42531913 | missense | NM_015559 | c.2608G>C | p.Gly870Arg | Pathogenic/COSM1684722 |
| 25 | C | ND | ND | C | ASXL1 | 20 | 31022288 | nonsense | NM_015338 | c.1773C>A | p.Tyr591* | Pathogenic/COSM1681609 |
| 25 | C | ND | ND | C | KIT | 4 | 55599321 | missense | NM_000222 | c.2447A>T | p.Asp816Val | Pathogenic/COSM1314 |
| 26 | C | ND | ND | C | GATA1 | X | 48649629 | missense | NM_002049 | c.113C>T | p.Pro38Leu | Likely Pathogenic/ COSM6498484, COSM6498483 |
| 27 | C | ND | ND | C | CALR | 19 | 13054627 | frameshift | NM_004343 | c.1154_1155insTTGTC | p.Lys385Asnfs* | Pathogenic/COSM1738056 |
| 27 | C | ND | ND | C | U2AF1 | 21 | 44514777 | missense | NM_001025203 | c.470A>G | p.Gln157Arg | Pathogenic/ COSM211532, COSM1724986 |
| 28 | C | ND | ND | C | ASXL1 | 20 | 31022441 | frameshift | NM_015338 | c.1934dupG | p.Gly646fs*12 | Pathogenic/COSM34210 |
| 28 | C | ND | ND | C | TP53 | 17 | 7577094 | missense | NM_000546 | c.844C>T | p.Arg282Trp | Pathogenic/COSM99925, COSM1636702, COSM10704 |
| 31 | C | ND | ND | C | FLT3 | 13 | 28608244 | inframe_24 | NM_004119 | c.1788_1811dup | p.Glu604_Phe605ins8 | Pathogenic/ITD |
| 31 | C | ND | ND | C | WT1 | 11 | 32417907 | frameshift | NM_000378 | c.1090_1093dupTCGG | p.Ala382ValfsTer4 | Pathogenic/COSM5487332 |
| 32 | C | ND | ND | C | CBL | 11 | 119148931 | missense | NM_005188 | c.1151G>A | p.Cys384Tyr | Pathogenic/COSM34066 |
| 32 | C | ND | ND | C | WT1 | 11 | 32417913 | frameshift | NM_000378 | c.1080_1087dupTCTTGTAC | p.Arg380LeufsTer72 | Likely Pathogenic/COSM5487152 |

P = Pan-Myeloid panel; M = Myeloid solutions panel; S = SureSeq panels; T = TruSight Myeloid Panel; Chr = chromosome; CodCons = coding consequence; C = Called; NI = Not Included; ND = Not Done; NC = Not Called; SNP = Single nucleotide polymorphism

relevant variants (mean = 1.5), followed by MYS (mean = 1.4), and SureSeq™ panels (mean = 1.1) (Fig 5C). There were 11 discordant variants, these variants were not detected because SureSeq and MYS did not include *GATA2*, *BCORL1*, *SH2B3* and *PTPN11* in their desing, hence mutations such as *GATA2* p.Ala318Gly and p.Ala318Val (patient 1 and 4), *BCORL1* p.Arg1048* and *SH2B3* p.Arg392Trp (patient 2), and *PTPN11* p.Gly60Cys (patient 3) could not be called. Similarly, SureSeq™ panels missed *SRSF2* p.Pro95His (patient 13) and *SF3B1* p.Lys666Asn (patient 15) variants because those genes were not included in its design. Patient 20, tested with TSMP and PMP, harbored the likely pathogenic mutation *GNAS* p.Arg844His, which was called by TSMP but not by PMP, again due to panel design.

ii. <u>Clinically relevant variants called in the core myeloid gene set</u>. A total of 37 clinically relevant variants fell in one of the 19 genes of the core myeloid gene set (Fig 5D, Table 3). All panels called the same variants, with the exception of 3 cases, for which SureSeq™ AML Panel did not call two *FLT3*-ITD variants p.Phe594_Arg595ins12, p.Tyr589_Phe590ins12 (patient 7 and 16) and SureSeq™ Core MPN Panel did not called one *CALR* p.Leu367 Thrfs*46 variant (patient 14). Of note, all three missed variants were indels with a length larger than 35bp. Additionally, 2 *FLT3*-ITD positive cases by conventional molecular techniques (patients 2 and 3) (Table 1), tested negative with the SureSeq™ AML, MYS and PMP NGS panels. Moreover, the insertion could not be visualized on the corresponding bam files within IGV, which means that the ITD- harboring alleles were either not captured during library preparation, or that the corresponding reads were not correctly aligned. These data suggest that NGS is prone to missing long indels.

**c) Comparison of all detected VAFs.** Correlation analysis between VAFs detected by each panel showed high level of concordance between SOPHiA GENETICS panels (S5A Fig and Fig 5A $R^2$ = 0,994) and acceptable concordance between SOPHiA GENETICS and Sure-Seq' panels (S5B and S5C Fig; $R^2$ = 0,953 and $R^2$ = 0,942, respectively). On the contrary, VAFs detected by TSMP and PMP showed an elevated level of dispersion (S5D Fig; $R^2$ = 0,767), indicating a relatively high discordance in detected VAF values between panels.

## Common sequencing errors detected in the NGS panels

Those variants with a VAF of < 5%, recurrently present in $\geq$ 30% of samples analyzed by any of the panels, and found within a repetitive region (homopolymeric regions or repeating triplets) defined as sequencing errors. We detected a total of 20 sequencing errors. Eight were present in 100% of the sequenced samples; 4 were called in more than one panel. Of note, TSMP was the panel that called a higher number of sequencing errors (n = 15), followed by PMP (n = 6), SureSeq™ AML panel (n = 3) and MYS panel (n = 2). Sequencing errors are listed in S6 Table.

## Discussion

Patients with MN are clinically heterogeneous. Mutations in the genes related with MNs are pathogenically important and confer a better understanding of the disease. Therefore, genetic testing might help clinicians choosing the best treatment for the patient, and predicting patient outcome. In this study we evaluated the utility of four targeted NGS gene panels (three commercially available and one custom), based on their technical features and clinicopathologic utility. The present analysis may offer helpful criteria to hematological genetic laboratories when implementing new NGS panels.

NGS panel target design, greatly depends on the intended use of the panel. Panels can be designed with a focus on a specific phenotype (e.g. AML or MDS with ring sideroblasts) or aiming to a wider scope (e.g. a pan-myeloid panel). In any case, a deep knowledge of the scientific literature of the disease of interest is necessary. Hence, we started our study by summarizing current information about all genes included in any of the four panels, and their relevance to MN (S5 Table).

All four panels had in common what we have called the "core myeloid gene set" of 19 genes, that have been extensively described in MN [2][35][36][37]. However, additional genes highly relevant to MN were not included in all four panels design: (i) *CBL*, *CSF3R*, *EZH2*, *PTPN11*, *SETBP1*, *SF3B1*, *SRSF2*, and *ZRSR2* genes were not included in SureSeq panels (Oxford Gene Technology, Oxford, UK) [26][38][39][40][41][42]; (ii) *BCOR*, *GATA1*, *KMT2A* and *PHF6* genes were not included in MYS panel (SOPHiA GENETICS, Saint Sulpice, Switzerland) [43][44][45][46]; (iii) TSMP and PMP were the only panels including exons from *ATRX*, *BCORL1*, *CUX1*, *GATA2*, *IKZF1*, *RAD21*, *SMC1A*, *SMC3*, and *STAG2* genes, all of them of interest in myeloid malignancies [43][47][48][49][50][51][52]. Interestingly, only PMP included *SH2B3* and *NF1* genes; *SH2B3* is highly expressed in hematological cells and its clinical relevance in MPNs has been described in several studies [53][54][55]; *NF1* mutations are thought to have a similar effect in leukemogenesis as mutations in the RAS pathway [25].

According to the literature, not all genes included in the panels have been shown to be clinically relevant. Therefore, when choosing an NGS panel, it might be important to prioritize the panel that includes all genes with diagnostic, prognostic and/or predictive value for the disease of interest. The clinical relevance of each gene included in all four panels is represented in Fig 2. The figure shows that *ABL1*, *CALR*, *MPL*, *JAK2* and *SF3B1* genes have diagnostic value, as described in several studies[2][18][37]. Similarly, *ABL1*, *CALR*, *JAK2*, *KIT*, *FLT3*, *IDH1* and *IDH2* gene mutations have FDA-approved treatments[56][57]. Patients harboring mutations in *TET2* and *DNMT3A* genes have been shown to present better response to hypomethylating agents [58][59]; *DNMT3* mutated patients could also benefit from daunorubicin induction therapy [60]. Fig 2 also shows a high number of genes related to prognosis, such as biallelic *CEBPA* and *SF3B1* (good prognosis), and *ASXL1* and *TP53* (poor prognosis) [9][10][61]. As mentioned above, not all panels included all genes with clinical relevance, and therefore, those panels would miss important information about patient outcome.

TSMP (Illumina, San Diego, CA, USA) has been extensively used on the study of myeloid malignancies [20][62][63]. However it faces a couple of challenges: firstly, the panel hampers the capture of GC regions (such in the case of *CEBPA*) because is based on amplicon technology; secondly, TSMP covered *ATRX* exon 11, that according to Illumina´s panel description it is not in the panel design; and finally, it included genes with clinical implications in lymphoid malignancies, like *CDKN2A* and *FBXW7* [64], *MYD88* [27][65][66][67], *NOTCH1* [28], and *PTEN* [29]. The fact that TSMP covered genes and regions not relevant to MN, might lessen the number of reads in the regions of interest. Of note, TSMP VCFs presented a high percentage (over 50%) of variants with a VAF of less than 5%, which might have been originated during PCR amplification [68]; this might also explain the divergent VAF between TSMP and the hybridization-based capture panels [69]. In addition, TSMP was the panel that showed more sequencing errors [70]. However, despite these issues, TSMP covered the majority of genes recurrently mutated in AML, MPN, MDS, and CMML, including all clinically relevant genes.

SureSeq™ panels (Oxford Gene Technology, Oxford, UK) were used combining two off-the-shelf panels available from OGT, designed for the study of AML and MPN, respectively. Currently, OGT also offers an extended MPN panel, but no wider myeloid solution panel was commercially available. Variant calling was done manually by their expert bioinformaticians, because their SureSeq™ Interpret Software was not available at the time of performing the

present study. This panel was the one showing lower coverage for all genes, probably due to the fact that all 16 samples were multiplexed on a V2 kit (8Gb per run; 150x2 cycles), whereas for the other three panels, 8 samples were multiplexed on a V3 kit (14Gb per run; 250x2 cycles); this might be the reason why *FLT3*-ITDs detected with low VAF in other panels, were not called with SureSeq™ AML panel. In contrast, it was the panel that called more variants within the core myeloid gene set, because the AML panel covered the CDS of all genes included. However, not all those covered extra regions have been reported as clinically relevant, and sequencing them lessens the read depth of the regions useful for clinical purposes. For example, out of the 12 exons of *IDH1* gene, only mutations in exon 4 have been reported as deleterious [71][72].

In this study, we have used two solutions from SOPHiA GENETICS: their commercially available MYS panel, and our custom PMP. PMP lacks three genes from MYS (*ABL1*, *BRAF*, and *HRAS*), but its larger design intends to be a pan-myeloid test, covering (i) genes related to sporadic MNs, (ii) genes described to confer a germline predisposition to MN, such as *ANKRD26*, *DDX41*, and *SRP72* (Fig 2, S5 Table)[73][74], and (iii) regions frequently affected by CNV, namely del (7q)/-7, del(5q), del(20q) and trisomy 8. Nevertheless, there is also room for improvement of PMP. For example: whole CDS of *ANKRD26* gene was covered, but 5' UTR should also be analyzed, since mutations related to disease progression are encompassed within 5´UTR through exon 2 [75][76]; and *FLT3* exons 11 and 13 are neither included in the panel design [34][77]. Of note, the other 3 panels did include exon 13, but only SureSeq panel included exon 11. Both MYS panel and PMP benefit from SOPHiA DDM software, which greatly facilitates variant classification.

In order to design or choose a commercially available panel, it is important to know the MN that it is going to be characterized. For instance, all four panels target genes for MPN, but PMP includes *MPL* exons 3, 4, 5 and 12 recently described as mutated in triple negative patients [78], whereas TSMP, SureSeq™ CoreMPN and MYS panels did not include those exons in their design. Moreover, TSMP and SureSeq™ CoreMPN panels did not cover *JAK2* exon 15, where mutations have been described [79]. PMP was designed in July 2017, which makes it the youngest of the four analyzed panels. This is probably the reason why its design is more up-to date with the literature. In fact, PMP is currently being upgraded, to fix *ANKRD26* and *FLT3* coverage, to target further genes related to predisposition to MN, and to include analysis of common rearrangements in myeloid disorders (through RNA sequencing) (e.g. *BCR-ABL1* for Chronic Myeloid Leukemia, *PML-RARA* for Acute Promyelocytic Leukemia, *etc.*). Actually, more recently available myeloid panels also include the study of translocations, such as Oncomine™ Myeloid Research Assay (ThermoFisher SCIENTIFIC, Waltham, MA, USA) and MYS+ panel (SOPHiA GENETICS, Saint Sulpice, Switzerland). It should be noted that Oncomine™ Myeloid Research Assay is an amplicon-based panel, and therefore it might face the same limitations as TSMP when it comes to GC-rich regions amplification; interestingly, it is the only one that includes gene expression testing.

In this project we have detected that any NGS panel is still facing, at least, two challenges in the myeloid field. On the one hand, the detection of indels: correct calling of ITDs in the fms-related tyrosine kinase 3 gene (*FLT3*-ITD) are crucial in AML, since they are associated to prognosis and to specific treatments [34][80]. In our cohort, two *FLT3*-ITD mutations of 36bp in length (detected by classical molecular techniques in our laboratory) were not called by any of the NGS gene panels tested in this study, which means that conventional diagnostics techniques are still essential for hematological malignancies diagnosis [81]. NGS difficulty for long *FLT3*-ITD detection has been reported before [62][82]; this is because current NGS chemistries employ short reading sequencing (read length 50-300bp) and this makes it prone to lose structural variants such as long indels [83][84]. In support of this observation, in our cohort, the three variants missed by SureSeq panels (sequenced at shorter read length than the other

panels, 150bp vs >200bp), were indels. On the other hand, molecular testing of CCAAT/enhancer binding protein A gene (*CEBPA*) is also crucial for patients with AML, as biallelic *CEBPA* is correlated with good prognosis [61]; however, those mutations fall usually one at C-terminal and the other one at the N-terminal region of the gene, so, again because of the short read issue, NGS technology cannot detect if the mutations fall in different alleles or in the same allele of the gene.

Besides the technical capacity of detecting variant types, when using NGS panels it is important to discriminate the clinically relevant variants from accompanying events. In our cohort, the number of pathogenic or likely pathogenic variants was two orders of magnitude smaller than the number of coding variants passing quality control (50 *vs* 1146). This drop highlights the importance of including expert geneticists familiar with hematological malignancies and NGS technology within the multidisciplinary genomic tumor board, as it has been suggested before [13][83].

In summary, based on the present study, the ideal NGS panel for the study of the myeloid malignancies should meet six requirements. (i) It should include in its design those genes described in MN to be clinically relevant for the pathology of the disease, being careful when choosing the relevant regions of each gene; this design requires periodical upgrade upon literature review. (ii) When studying SNV and indels, the chemistry should enable capturing all relevant genomic regions; hybridization capture-based panels usually evade the GC-rich regions glitches of an amplicon-based panel. (iii) It should have the capacity of detecting long indels, which is particularly important when it comes to defy the *FLT3*-ITD detection challenge. (iv) Since sequencing costs are gradually decreasing, genetic laboratories' dream is that NGS technology provides a "just one test" for all relevant genetic abnormalities contemplated in WHO and European LeukemiaNet (ELN) guidelines [2][80]; therefore the ideal myeloid NGS panel should be able to simultaneously analyze SNVs, indels, CNVs, aberrant gene expression, and common gene rearrangements. (v) The turnaround time (TAT) for reporting should comply with current ELN guidelines [80]. For example, TAT for *NPM1* and *FLT3* reporting is 48–72 hours; however, sample processing, NGS library preparation, sequencing and reporting, take a minimum of 4 working days, which means that, for now, conventional molecular testing needs to be kept in place. (vi) Sequencing data should be interpreted by two geneticists, at least one of them with expertise in hematological malignancies, and both of them familiar with the challenges inherent to NGS technology [83].

## Conclusion

The current study describes the performance of four NGS panels focused on MN from the technical and clinical perspective. Our data show that there is a risk of finding different mutations depending on the panel of choice. This discordance is motivated by panel design and sequencing data analysis. MN are genetically heterogeneous, therefore choosing a commercial NGS panel needs detailed study of its scope, to be aware of its limitations and to avoid missing the testing of genes relevant to a specific MN subtype.

Based on our data, the characterization of some genetic regions (*CEBPA*, *CALR*, and *FLT3)* remains a challenge for NGS; this is a major issue, since AML and MPN management strongly depends on their correct detection. In addition, NGS testing times are hard to harmonize with TAT established in current ELN guidelines. Therefore, conventional molecular testing might need to be kept in place for the correct diagnosis of MN in some instances for now.

## Supporting information

**S1 Fig. Detail of target region for genes differing between panels.** SureSeq panels design included a larger target region of *ASXL1*, *FLT3*, *IDH1*, *IDH2*, *KIT*, *KRAS*, *NPM1*, *NRAS*, *TET2*, *U2AF1* and *WT1* genes, whereas *JAK2* gene was more widely covered by MYS panel, and *MPL*

gene by PMP.
(TIF)

**S2 Fig. Panel scope by genetic region.**
(TIF)

**S3 Fig. *CEBPA* gene coverage in all four NGS panels.** IGV screenshot showing genomic position (top track), *CEBPA* gene structure (bottom track) and coverage for the different panels (four central tracks). Panel tracks show differential coverage in grey color, and reads 1 and 2 in red and blue bars. TSMP track shows poor and heterogeneous coverage for *CEBPA* gene.
(TIF)

**S4 Fig. Panel coverage by genetic region.**
(TIF)

**S5 Fig. Comparison of the detected variants' VAF. A:** Comparison between variants called by PMP and MYS panel in their 27 genes in common. **B:** Comparison between variants called by PMP and SureSeq panels in their 23 genes in common. **C:** Comparison between variants called by MYS and SureSeq panels in their 19 genes in common. **D:** Comparison between variants called by PMP and TSMP in their 40 genes in common.
(TIFF)

**S1 Table. TruSight Myeloid Panel (TSMP) target regions per gene.** TSMP includes a total of 54 genes for SNV and indels.
(DOCX)

**S2 Table. Myeloid Solutions Panel (MYS) target regions per gene.** MYS panel design includes a total of 30 genes for SNV and indels.
(DOCX)

**S3 Table. SureSeq panel target regions per gene.** SureSeq™ AML panel design includes a total of 20 genes and SureSeq™ CoreMPN panel design includes 3 genes for SNV and indels.
(DOCX)

**S4 Table. Pan Myeloid Panel (PMP) target regions per gene.** PMP panel design includes a total of 48 genes for SNV and indels.
(DOCX)

**S5 Table. Frequency of gene mutations in myeloid malignancies.**
(DOCX)

**S6 Table. Common sequencing errors detected in the NGS gene panels.**
(DOCX)

## Acknowledgments

This work was funded by the Government of Navarra, Department of Industry, Energy and Innovation (Project DIANA, 0011-1411-2017-000028); and supported by CIMA LAB Diagnostics research program.

We are grateful to Oxford Gene Technology team, especially David Cook, and SOPHIA GENETICS team, especially José Maria Belloso, for technical assistance and fruitful discussions.

AAD is supported by a CIMA´s fellowship; MFM and her research is supported by the Spanish Association against Cancer (AECC, AIO2014) and ISCIII (Ministerio de Economía y

Competitividad of Spanish central government, PI16/00159); IV is supported by Pethema; FP aknowledges funding from ISCIII (PI17/00701).

And finally, we are particularly grateful to the patients who have participated in this study, and to the Biobank of the University of Navarra for its collaboration.

## Author Contributions

**Conceptualization:** Ramón García-Sanz, María Isabel Prieto-Conde, María del Carmen Chillón, Marta Fernandez-Mercado, María José Calasanz.

**Data curation:** Almudena Aguilera-Diaz, Iria Vazquez, Beñat Ariceta, Amagoia Mañú, Zuriñe Blasco-Iturri, María José Larrayoz, Ana Alfonso-Pierola.

**Formal analysis:** Almudena Aguilera-Diaz, Beñat Ariceta.

**Funding acquisition:** Felipe Prosper, Marta Fernandez-Mercado, María José Calasanz.

**Investigation:** Almudena Aguilera-Diaz, Iria Vazquez, María José Larrayoz.

**Methodology:** Almudena Aguilera-Diaz, Amagoia Mañú, Zuriñe Blasco-Iturri, Sara Palomino-Echeverría, Marta Fernandez-Mercado.

**Project administration:** Iria Vazquez.

**Resources:** Felipe Prosper, Marta Fernandez-Mercado.

**Software:** Beñat Ariceta.

**Supervision:** Iria Vazquez, Felipe Prosper, Marta Fernandez-Mercado, María José Calasanz.

**Validation:** Ramón García-Sanz, María Isabel Prieto-Conde, María del Carmen Chillón.

**Writing – original draft:** Almudena Aguilera-Diaz, Marta Fernandez-Mercado.

**Writing – review & editing:** Iria Vazquez, Beñat Ariceta, Amagoia Mañú, Zuriñe Blasco-Iturri, Sara Palomino-Echeverría, María José Larrayoz, Ramón García-Sanz, María Isabel Prieto-Conde, María del Carmen Chillón, Ana Alfonso-Pierola, Felipe Prosper, María José Calasanz.

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
