## [Decision Letter · Decision Letter 0]

18 Sep 2019

PONE-D-19-20399

Assessment of the clinical utility of four NGS panels in myeloid malignancies. Suggestions for NGS panel choice or design

PLOS ONE

Dear Dr Fernandez-Mercado,

Thank you for submitting your manuscript to PLOS ONE. After careful consideration, we feel that it has merit but does not fully meet PLOS ONE’s publication criteria as it currently stands. Therefore, we invite you to submit a revised version of the manuscript that addresses the points raised during the review process.

We would appreciate receiving your revised manuscript by Nov 02 2019 11:59PM. To enhance the reproducibility of your results, we recommend that if applicable you deposit your laboratory protocols in protocols.io, where a protocol can be assigned its own identifier (DOI) such that it can be cited independently in the future. For instructions see: http://journals.plos.org/plosone/s/submission-guidelines#loc-laboratory-protocols

We look forward to receiving your revised manuscript.

Kind regards,

Honey V. Reddi

Academic Editor

PLOS ONE

Journal Requirements:

2. In the ethics statement in the manuscript and in the online submission form, please provide additional information about the patient samples and records retrieved from the biobank. Specifically, please ensure that you have discussed whether all data were fully anonymized before you accessed them and/or whether the IRB or ethics committee waived the requirement for informed consent. If patients provided informed written consent to have data from their medical records used in research, please include this information.'

3. Thank you for including your ethics statement: "Samples and data from patients included in the study were provided by the Biobank of the University of Navarra (UN) and were processed following standard operating procedures approved by the Ethical and Scientific Committees of the UN."

Additional Editor Comments:

The Authors have put together a timely study. Unfortunately this manuscripts suffers from a lack of clarity in terms of design and write-up including presentation of the results. This paper would require major revision in order to be considered for publication.

Major points:

1. Why were a select set of samples not tested across all panels? There needs to be an exact comparison to provide credence to their hypothesis.

2. What was the reason for extra samples (total of 32) only being tested on the in-house panels.

3. The results should be clearly iterated as well as all points specifically addressed in the discussion.

Minor Points:

1. The manuscript needs to be edited for typos, grammar and language.

Reviewers' comments:

Reviewer's Responses to Questions

**Comments to the Author**

1. Is the manuscript technically sound, and do the data support the conclusions?

Reviewer #1: Yes

2. Has the statistical analysis been performed appropriately and rigorously? 

Reviewer #1: Yes

3. Have the authors made all data underlying the findings in their manuscript fully available?

Reviewer #1: Yes

4. Is the manuscript presented in an intelligible fashion and written in standard English?

Reviewer #1: Yes

5. Review Comments to the Author

Reviewer #1: In this manuscript, the authors investigated four NGS panels in myeloid malignancies. They evaluated three commercially available targeted NGS gene panels and one custom panel based on the technical features and variant detection of myeloid neoplasms related genes. The manuscript is well written. Below are the comments I have:

1, The authors collected total 32 patient samples: 17 with AML, 7 with MPN, 6 withMDS, and 2 with CMML in this study. All 32 samples were tested with the custom panel. However, only some of the samples were used for the other three commercial NGS panels. Was there any reason that the authors did not include all 32 samples for testing all four panels? It would be clearer if the authors could provide information on how the samples were selected for each panel testing and what sample types were included.

2, In Table 2, the authors listed the software used for different panel analysis. However, the analysis tool for the custom panel was developed in-house. At the time of this study, was this tool validated for variant calling?

3, The clinical relevance of each gene included in all four panels has been represented in Figure 2. However, I found it difficult to understand the label of Figure 2. It is unclear what the clinical relevancies are for. Is it based on the summary of different clinical relevance categories results?

4, The authors discussed that the sequencing errors of TSMP were likely due to its amplicon-based nature. However, with only one amplicon-based targeted panel included in the comparison, this conclusion is not convincing. In addition, the analysis software of TSMP was different from other panels. It is possible that the variants calling is affected by the software.

5, The authors listed 19 genes as the core myeloid gene set. Four references papers were cited in the discussion to support this core gene set. However, it is unclear that how these 19 gene were selected since there were other genes listed in all four papers.

6. PLOS authors have the option to publish the peer review history of their article (what does this mean?). If published, this will include your full peer review and any attached files.

Reviewer #1: No

---

## [Author Response · Author response to Decision Letter 0]

18 Oct 2019

To the Editor

Thank you for pointing out the need of amending the ethics statement. Now it reads as follows: “Samples and data from patients included in the study were provided by the Biobank of the University of Navarra (UN) and were processed following standard operating procedures approved by the CEI (Comité de Ética de la Investigación) of UN. Patient’s data were fully anonymized, and all patients provided informed written consent to have data from their medical records such as age, gender and diagnosis to be used for research purpose.”

The Ethics Statement has also been updated in the submission form.

1. We agree with the Editor that all samples should have been tested across all panels. Unfortunately, we could not run the 32 samples with the four NGS gene panels, due to limited sample availability. We have included in the study only the 32 cases with enough DNA to build libraries with at least two different panels. 

2. Our study includes 3 commercially available panels and only 1 custom (in-house) panel. We have included in the study 32 cases analyzed with at least two different panels. Figure 1 shows which panels (2, 3 or 4) have been used to sequence each sample. 

3. We thank the Editor for suggesting to iterated the results. We think that the manuscript now reads more clear and it is easier to follow the conclusions.

We have now carefully reviewed the full manuscript and all typos and grammar issues have been addressed. 

To the Reviewer #1

1. The reason why it was not possible to test the 32 samples with all four panels, was material constrains. We have included in the study those cases with enough DNA to build libraries with at least two different panels. When in doubt, what panel to use in a particular sample, we prioritized library construction with the panel which covered previously known variants in a particular sample. 

2. Raw data from the custom panel were uploaded onto SOPHiA DDM software, where alignment, variant calling and annotation were performed. SOPHiA Genetics develops custom panels including a panel validation program: firstly, the company performs the panel technical validation in their facilities; secondly, clinical validation is performed in the hospital facilities, including samples with multiple known mutations, aiming at confirming the capability of the SOPHiA DDM software of calling those variants.

3. We agree with the Reviewer that Figure 2 legend was confusing. We have now rewritten the Figure legend. It explains that clinical relevance of the genes refers to their diagnostic, prognostic and/or predictive value, or their relation to predisposition to develop MN. The legend now reads as follows:

“Figure 2. Genes covered by each panel and their clinical relevance. The 62 genes included in the present study are listed on the right. Black color denotes which gene is covered in each panel. Green color highlights the 53 genes that have been described as clinically relevant for MN, since they show diagnostic, prognostic and/or predictive value, or they have been related to predisposition to develop MN. Red color represents genes that are not clinically relevant in MN. Grey color marks those genes that has been described in MN but their clinical relevance is still unknown”

4. We agree with the Reviewer that only one amplicon-based targeted panel is not enough to conclude anything about amplicon based-panels. Hence, we have now deleted the sentence in which we speculated about the reason of abundant sequencing errors in this panel.

5. We thank the Reviewer for raising this point. We have named the 19 genes which are included in all four panels “core myeloid gene set” for the sake of clarity in the analysis. It is only a descriptive term; we do not mean that those are the only genes to be sequenced in MN. In fact, based on an extended literature research that we carried out, we mention in the manuscript the importance of updating the list of genes with clinical relevance in MN.

---

## [Decision Letter · Decision Letter 1]

21 Nov 2019

PONE-D-19-20399R1

Assessment of the clinical utility of four NGS panels in myeloid malignancies. Suggestions for NGS panel choice or design

PLOS ONE

Dear Dr Fernandez-Mercado,

Thank you for submitting your manuscript to PLOS ONE. After careful consideration, we feel that it has merit but does not fully meet PLOS ONE’s publication criteria as it currently stands. Therefore, we invite you to submit a revised version of the manuscript that addresses the points raised during the review process.

We would appreciate receiving your revised manuscript by Jan 05 2020 11:59PM. To enhance the reproducibility of your results, we recommend that if applicable you deposit your laboratory protocols in protocols.io, where a protocol can be assigned its own identifier (DOI) such that it can be cited independently in the future. For instructions see: http://journals.plos.org/plosone/s/submission-guidelines#loc-laboratory-protocols

We look forward to receiving your revised manuscript.

Kind regards,

Honey V. Reddi

Academic Editor

PLOS ONE

Additional Editor Comments (if provided):

We thank the authors for responding to our comments and making the necessary edits. While the technical content of the manuscript is now sound, there still needs to be some clarity and flow to the manuscript to ensure the message is clear. For example in line 41, you refer to NGS as a fundamental instrument, which is not accurate. NGS is a technology. It would help get the message across if the flow of the manuscript is a bit smoother.

Reviewers' comments:

Reviewer's Responses to Questions

**Comments to the Author**

1. If the authors have adequately addressed your comments raised in a previous round of review and you feel that this manuscript is now acceptable for publication, you may indicate that here to bypass the “Comments to the Author” section, enter your conflict of interest statement in the “Confidential to Editor” section, and submit your "Accept" recommendation.

Reviewer #1: All comments have been addressed

2. Is the manuscript technically sound, and do the data support the conclusions?

Reviewer #1: Yes

3. Has the statistical analysis been performed appropriately and rigorously? 

Reviewer #1: Yes

4. Have the authors made all data underlying the findings in their manuscript fully available?

Reviewer #1: Yes

5. Is the manuscript presented in an intelligible fashion and written in standard English?

Reviewer #1: Yes

6. Review Comments to the Author

Reviewer #1: (No Response)

7. PLOS authors have the option to publish the peer review history of their article (what does this mean?). If published, this will include your full peer review and any attached files.

Reviewer #1: No

---

## [Author Response · Author response to Decision Letter 1]

20 Dec 2019

Editor Comments

We thank the authors for responding to our comments and making the necessary edits. While the technical content of the manuscript is now sound, there still needs to be some clarity and flow to the manuscript to ensure the message is clear. For example in line 41, you refer to NGS as a fundamental instrument, which is not accurate. NGS is a technology. It would help get the message across if the flow of the manuscript is a bit smoother.

We thank the Editor for raising this point, which has now been taken under consideration. We have adapted the manuscript according to the recommendations, and we believe that the paper now flows smoother.

Reviewer#1 

There are no additional points raised by Reviewer 1

---

## [Editor Report · Decision Letter 2]

6 Jan 2020

Assessment of the clinical utility of four NGS panels in myeloid malignancies. Suggestions for NGS panel choice or design

PONE-D-19-20399R2

Dear Dr. Fernandez-Mercado,

We are pleased to inform you that your manuscript has been judged scientifically suitable for publication and will be formally accepted for publication once it complies with all outstanding technical requirements.

With kind regards,

Honey V. Reddi

Academic Editor

PLOS ONE
---

## [Editor Report · Acceptance letter]

13 Jan 2020

PONE-D-19-20399R2 

Assessment of the clinical utility of four NGS panels in myeloid malignancies. Suggestions for NGS panel choice or design 

Dear Dr. Fernandez-Mercado:

I am pleased to inform you that your manuscript has been deemed suitable for publication in PLOS ONE. Congratulations! Your manuscript is now with our production department. 

With kind regards,

on behalf of

Dr. Honey V. Reddi 

Academic Editor

PLOS ONE